# Epigenetic Regulation Interplays with Endometriosis Pathogenesis in Low-Birth-Weight Patients via the Progesterone Receptor B–VEGF-DNMT1 Axis

**DOI:** 10.3390/diagnostics13122085

**Published:** 2023-06-16

**Authors:** Arief Setiawan, Ruswana Anwar, Mas Rizky Anggun Adipurna Syamsunarno, Johanes Cornelius Mose, Budi Santoso, Ani Melani Maskoen, Wiryawan Permadi, Budi Setiabudiawan, Meita Dhamayanti, Yudi Mulyana Hidayat

**Affiliations:** 1Endocrinology Reproduction and Fertilisation Division, Department of Obstetrics and Gynaecology, Hasan Sadikin Hospital, Bandung 40161, Indonesia; 2Faculty of Medicine, Universitas Padjadjaran, Bandung 40161, Indonesia; 3Faculty of Medicine, Universitas Airlangga, Surabaya 60132, Indonesia

**Keywords:** epigenetic, endometriosis, low birth weight, progesterone receptor B, DNA methyltransferase, VEGF

## Abstract

Background: Low birth weight (LBW) is a risk factor associated with endometriosis. Our study aimed to analyze the risk of endometriosis in women with a LBW history and the relationships of progesterone receptor B (PR-B) gene promoter methylation, DNA methyltransferase-1 (DNMT1) expression, PR-B expression, and vascular endothelial growth factors (VEGF) with endometriosis. Methods: This study was conducted in two stages, a retrospective case-control design and a cross-sectional design, with 52 cases of endometriosis and 30 controls, which were further subdivided into LBW and non-LBW groups, at Hasan Sadikin General Hospital and its hospital networks from October 2017 to August 2021. Menstrual blood was taken from subjects and analyzed using pyrosequencing techniques to assess DNA methylation, while q-RT PCR was used to assess gene expression. Results: There were significant differences in PR-B methylation, DNMT1 expression, PR-B expression, and VEGF expression (*p* < 0.001) between the case and control groups. There was a significant negative correlation between PR-B methylation and PR-B expression (r = −0.558; *p* = 0.047). Based on a multiple logistic analysis, the most dominant factor affecting endometriosis incidence is PR-B (OR 10.40, 95% CI 3.24–33.4, R^2^ = 45.8). We found that patients with a low birth weight history had a 1.41-times-higher risk of developing endometriosis (95% CI 0.57–3.49, *p* = 0.113), although the relationship was not statistically significant. Conclusion: Endometriosis is associated with PR-B gene promoter hypermethylation, decreased PR-B expression, and increased DNMT1 and VEGF expression. The methylation of PR-B is the most dominant factor affecting endometriosis incidence.

## 1. Introduction

Endometriosis is a gynecological disease often found in women of reproductive age. Genetic and environmental factors play major roles in endometriosis etiopathogenesis, and, nowadays, endometriosis is considered an epigenetic disease. LBW has been recognized as a risk factor for developing endometriosis in later life [1,2,3,4,5]. This risk occurs during the fetal programming phase inside the womb and is related to an unsupportive intrauterine environment, namely fetal malnutrition. This malnutrition state forces the fetus to adapt to this unsupportive environment, resulting in various metabolic and endocrine function disorders, that eventually cause LBW infants to be more susceptible to developing immuno-inflammatory disorders later in life, including endometriosis [5,6,7,8,9].

Women with an LBW history have a greater tendency to suffer from endometriosis in later life. This is presumed to be related to aberrations in DNA methylation correlated with epigenetic processes [5,8,9]. Endometriosis is known to be associated with the hypermethylation of the PR-B promoter gene. PR-B hypermethylation stimulates a feedback response causing a down-regulation of progesterone receptor B that eventually leads to progesterone resistance. In contrast, PR-A has minimal activity and opposes the transcriptional activity of PR-B. PR-A and PR-B are thought to function as competing systems for target cells to control progesterone responsiveness. Progesterone signaling is enhanced in a PR-B-dominant condition, whereas progesterone responsiveness is decreased in a PR-A-dominant state [10]. Moreover, it has also been previously reported that DNA methyltransferase-1 (DNMT1), an enzyme involved in DNA methylation, is overexpressed in endometriosis [10,11]. With progesterone resistance occurring, the level of 17βHSD type 2 decreases, causing an increase in estradiol levels that eventually lead to growth of endometriosis [12]. Progesterone resistance and the uncontrolled increase of the estradiol level in endometrial tissue also initiates an increase of vascular endothelial growth factor (VEGF) that causes angiogenesis and neurogenesis in endometriosis. All of the above mechanisms lead to the development and progression of endometriosis [4,13,14].

Referring to Sampson theory, during the menstrual phase, it is very feasible to obtain an endometrial cell sample from menstrual blood. Menstrual blood itself contains DNA with the same methylation status as eutopic endometrial tissue, and therefore can be regarded as a non-invasive medium to help diagnosing endometriosis [15].

To the best of our knowledge, there has been no other study that retrospectively investigated the association of LBW history with endometriosis risk in later life through analysis of its relationship with PR-B receptor gene promoter methylation, DNMT1 expression, PR-B receptor expression, and VEGF expression. All of these factors can be considered as evidence of epigenetic mechanisms in women with an LBW history and risk of endometriosis. We hope that the results of this study may provide new preventive efforts against endometriosis in terms of epigenetics.

## 2. Materials and Methods

### 2.1. Study Design and Eligibility Criteria

This study was conducted in two stages, namely a retrospective case-control study to retrospectively evaluate LBW history among women with endometriosis and controls, and a cross-sectional design to analyze the methylation of the PR-B receptor gene promoter, DNMT1 expression as an enzyme that mediates DNA methylation, PR-B receptor expression, and VEGF expression. Women of reproductive age (25–45 years old), history of term pregnancy with or without LBW, and regular menstrual cycles in the last 3 cycles were included in this study, between October 2017 and August 2021. Those with diagnostic results showing endometriosis were designated as the case group while those with diagnostic results showing no endometriosis were included in the control group. Exclusion criteria in this study were those who received hormonal therapy for the last three months, those with clinical suspicion of a pelvic infection, and those with malignancy both clinically and histopathologically.

Written informed consent was obtained from all subjects before the study. Those who agreed were asked to fill in a questionnaire on biodata, gynecological, and obstetric history. The subjects were then asked to come to a gynecology or endocrine clinic during their first or second day of menstruation. The questionnaires used were made by the study team and had not been pilot-tested beforehand. The birth weight data were obtained retrospectively from medical records. A low birth weight (LBW) was defined as a birth weight of less than 2500 g. If the information was not available, it was taken from interviews of the patients or their families.

### 2.2. Sample Acquisition

The first day of menstruation was defined as 24 h after the first realization of menstruation. Menstrual blood was taken from the posterior fornix using a 3 mL syringe and then was put into an EDTA tube. If, at the time of inspection, no blood was visible in the posterior fornix, the sample was taken using manual vacuum aspiration with a Karman cannula no. 4, 5, or 6.

### 2.3. Sample Processing

Assessment of the PR-B promoter methylation level was conducted by pyrosequencing with DNA samples, while examination of the DNMT1, PR-B and VEGF mRNA expression was conducted using quantitative real-time polymerase chain reactions (qRT-PCR).

Pyrosequencing involves a cascade of enzymes and specific substrates that produces a light signal upon nucleotide incorporations and subsequent pyrophosphate release. The light signal is detected via a charge-coupled camera to be converted into a quantitative electrical signal. The introduction of a sodium bisulfite treatment of the DNA samples before the pyrosequencing process enables the assessment of the methylation level of a given DNA segment. The bisulfite treatment results in the hydrolytic deamination of nonmethylated cytosines to uracils, which are subsequently replaced by thymines during PCR amplification. Methylated cytosines are resistant to conversion under the applied conditions. Thus, the degree of methylation at a given segment is calculated as the proportion of cytosines (former methylated cytosine) to all pyrimidines (cytosines + thymines). MS-HRM (methylation-specific high-resolution melting) was used to determine the methylation level of the samples [16].

The isolation of RNA from the endometrial cells of menstrual blood was carried out automatically using a MagNA Pure LC 2.0 machine from Roche and using a MagNA Pure LC RNA Isolation Kit III (Tissue) reagent. The gene expressions for DNMT1, PR-B, and VEGF are presented as the relative change in gene expression normalized to Glyceraldehyde 3-phosphate dehydrogenase (GADPH) as the reference/housekeeping gene. This study used PyroMark Gold Q96 Reagents and Pyromark Q96 Plate Low for the pyrosequencing process.

The primers for GADPH were 5′-GAA GGT GAA GGT CGG AGTC-3′ (forward) and 5′-GAA GAT GGT GAT GGG ATT TC-3′ (reverse); for DNMT1 they were 5′-AACCTTCACCTAGCCCCAG-3′ (forward) and 5′-CTCATCCGATTTGGCTCTTCA-3′ (reverse); for PR-B they were 5′-GGC AGA TGC TGT ATT TTG CACC-3′ (forward) and 5′-CAA ACC AAT TGC CTT GAT GAG-3′ (reverse); and for VEGF they were 5′-GCT ACT GCC ATC CAA TCD AG-3′ (forward) and 5′-CT CT CCT ATG TGC TGG CCTT-3′ (reverse).

#### 2.3.1. DNA Isolation

The isolation of DNA from the endometrial epithelial tissue of menstrual blood was carried out using the QIA & DNA mini kit protocol, as described below:The blood sample was removed from the refrigerator where it was stored, and then 180 µL of ATL buffer was added.An amount of 20 µL of proteinase K was added, and the mixture was homogenized by vortexing for 10 s. It was then incubated at 56 °C until the tissue was completely lysed.The liquid in the lid of the tube settled to the bottom as it was centrifuged at 5000 rpm for 10 s.An amount of 4 µL of RNase A (100 mg/mL) was added, and the mixture was centrifuged for 15 s. It was then incubated for 2 min at room temperature. The liquid on the lid was briefly spun to precipitate before adding 200 µL of AL buffer to the sample. Mixing again using a vortex for 15 s, it was incubated at 70 °C for 10 min. After incubation was complete, the 1.5 mL Eppendorf tube was spun for 10 s at 5000 rpm.A total of 250 µL of ethanol (96–100%) was added to the sample and centrifuged for 15 s. It was then centrifuged at 5000 rpm for 10 s.The solution was carefully transferred into the QIA amp mini spin column (in a 2 mL tube) without wetting the edges. The tube was closed and rotated at 6000 g for 1 min. The QIA amp mini spin column was placed in a clean 2 mL tube (supplied), and the tube containing the filtrate was discarded.An amount of 500 µL of AW1 buffer was added. The tube was closed and rotated at 6000 g for 1 min. The QIA amp spin column was placed in a clean 2 mL tube, and the tube containing the filtrate was discarded.A total of 500 µL of AW2 buffer solution was added, and it was rotated at full speed (20,000 g) for 3 min.The QIA amp mini spin column was placed in a new 2 mL tube, and the old tube containing the filtrate was discarded. It was spun at full speed for 1 min.The QIA amp spin column was placed in a clean 1.5 mL Eppendorf tube, and the tube containing the filtrate was discarded. A total of 200 µL of AE buffer or distilled water was added. It was incubated at room temperature for 1 min and then spun at 6000 g for 1 min.

#### 2.3.2. Bisulfite Addition

This protocol is based on Qiagen’s EpiTec Bisulfite. The procedure was as follows:The DNA sample was removed from refrigerator storage. The bisulfite mix was dissolved by adding 800 µL of RNase-free water (RNAse-free water).The bisulfite reaction was prepared in a 200 µL PCR tube.The PCR tube was closed, and the bisulfite reaction was thoroughly mixed. The tube was stored at room temperature (15–25 °C).The bisulfite DNA was changed using a PCR thermal cycler machine (this process took about 5 h).The PCR tube containing the bisulfite reaction was placed on the thermal cycler. Incubation with the thermal cycler was started.After the bisulfite conversion was complete, the PCR tube containing the bisulfite reaction was spun, and the finished bisulfite reaction was transferred into a sterile 1.5 mL Eppendorf tube.An amount of 560 µL of BL buffer, which already contained 10 g/mL of carrier RNA, was added. The solution was mixed, and then rotated briefly.The EpiTec spin column tube and collection tube were placed in the appropriate racks. All material from the tube in step 7 was transferred into the EpiTec spin column.The spin column was rotated at maximum speed for 10 s. The remaining filtrate was discarded, and the spin column was returned to the previous tube.A total of 500 µL of BW wash buffer was added to each spin column, and it was rotated at maximum speed for 1 min. The remaining filtrate was discarded, and the spin column was placed into the previous tube.An amount of 500 µL of DB buffer (desulphonation buffer) was added to each spin column, and it was incubated for 15 min at room temperature (15–25 °C).The spin column was rotated at maximum speed for 1 min. The remaining filtrate was discarded, and the spin column was placed into the previous tube.A total of 500 µL of BW washing buffer was added to each spin column, and it was spun at maximum speed for 1 min. The remaining filtrate was discarded, and the spin column was placed into the previous tube.Step 13 was repeated.The spin column was placed into the 2 mL tube, and the column was spun at maximum speed for 1 min to remove any remaining liquid.The spin column with the lid open was placed into a clean 1.5 mL microtube, and the column was incubated for 5 min at 56 °C.The spin column was placed into a sterile 1.5 mL micro four tube. An amount of 20 µL of EB buffer was added to the center of each membrane. The pure DNA was dissolved by spinning for 1 min at 15,000 g.

#### 2.3.3. RT-PCR

This process was used for the analysis of the DNMT1 gene expression. RNA was taken from the endometrial epithelial tissue of menstrual blood. This process used a KAPA SYBR FASR reagent, a forward primer (10 µM), a reverse primer (10 µM), KAPA RT Mix (50×), nuclease-free water, and an RNA template (1 pg−100 pg). The procedure was as follows:An amount of 40 mg of sample tissue was homogenized using liquid nitrogen and a mortar that had been soaked in DEPC.Then the sample was placed into a microtube containing 1 mL of RNA lysis buffer. A total of 175 µL of lysate was transferred into a new microtube, and 350 µL of dilution buffer RNA solution was added. The samples were incubated at 70 °C for 3 min. After incubation, the samples were incubated again for 10 min at 13,000 rpm.The resulting supernatant was then transferred to a new microtube containing 200 µL of 95% ethanol. The samples were transferred to the spin basket assembly and rotated for 1 min at 13,000 rpm.The eluent was removed, and 600 µL of RNA wash solution was added. The samples were spun again for 1 min at a speed of 13,000 rpm.A DNAse incubation mix (40 µL yellow core buffer, 5 µL DNAse 1, 5 µL MnCl_2_) was then added to the samples and they were incubated for 15 min at room temperature. After the incubation period was complete, 200 µL of DNAse stop solution was added, and the mixture was rotated for 1 min at 13,000 rpm.A total of 600 µL of RNA wash solution was added to the samples and they were rotated for 1 min at 13,000 rpm. The resulting lysate was discarded, and then 250 µL of RNA wash solution was added and rotated for 2 min at 13,000 rpm.The lysate and collection tube were discarded, and a new microtube was attached to the spin basket. Then, 100 µL of nuclease-free water was added and rotated for 1 min at 13,000 rpm.The obtained RNA was then measured using a spectrophotometer at wavelengths of 260, 280, and 320 nm to determine the concentration.

The cDNA synthesis was performed using the BioRad iScript cDNA synthesis kit. The reaction mixture was incubated for 5 min at 25 °C, then 30 min at a temperature of 42 °C, and 5 min at 85 °C. The DNMT1 mRNA expression was determined using relative quantitative real-time PCR. PCR reactions were carried out using a real-time PCR detection system (BioRad Laboratories Inc., Hercules, CA, USA). The cycle used is incubation at 95 °C for 3 min followed by 39 cycles of denaturation at 95 °C for 15 s, annealing at 57 °C for 20 s and extension at 72 °C for 25 s.

### 2.4. Statistical Analysis

The significance level of this study was set at 5%, with a power of 80%, and the magnitude of the standard deviation of birth weight was set at 540 g (for the case/endometriosis group) and 474 g (for the control/non-endometriosis group). The difference in mean birth weight between case and control groups was expected at 300 g. Therefore, a minimum sample size of 50 was required for the endometriosis group and 27 for the non-endometriosis group [17].

Data analysis was carried out using the SPSS software for Windows (version 13.0; SPSS, Inc., Chicago, IL, USA). Comparative analysis was performed using an unpaired *t*-test or Mann–Whitney test, while correlation analysis was conducted using a Pearson correlation test or Spearman test, with *p* < 0.05 considered significant and *p* < 0.01 considered highly significant.

## 3. Results

There were 52 subjects in the case group and 30 subjects in the control group. The baseline characteristics of the subjects are presented in Table 1.

Case and control subjects included in this study were then evaluated retrospectively for a history of LBW and classified based on it. There were 27 subjects who had a history of LBW (birth weight < 2.500 g) in the case group, while the remaining 25 had a birth weight of ≥2.500 g. On the other hand, in the control group, there were 13 subjects with an LBW history and 17 subjects without an LBW history.

Using the chi-square test, we found that patients with a history of LBW had a 1.41-times-higher risk of developing endometriosis compared to those without a history of LBW, although the relationship was not statistically significant (95% CI 0.57–3.49, *p* = 0.113). There was also no significant relationship found between history of LBW and endometriosis incidence (*p* = 0.054).

In Table 2 we can see that when compared with the other three groups (endometriosis without an LBW history, non-endometriosis with an LBW history, and non-endometriosis without an LBW history), endometriosis patients with an LBW history had the highest median of PR-B methylation, DNMT1 expression, PR-B receptor expression, and VEGF expression (*p* < 0.001).

We also used the Spearman rank test to analyze the correlation coefficient for the four variables studied in the four different sub-groups. We found a significant positive correlation between PR-B methylation with DNMT1 expression in the non-endometriosis group with a history of LBW (r = 0.572; *p* = 0.041), meaning that the higher the PR-B methylation is the higher the DNMT1 expression will be. Furthermore, in non-endometriosis subjects with an LBW history, we also found significant negative correlations (r = −0.797; *p* = 0.001) between PR-B expression and DNMT1 expression; between DNMT1 expression and VEGF expression (r = 0.616; *p* = 0.025); and, also, between PR-B methylation and PR-B expression (r = −0.558; *p* = 0.047).

To evaluate the potential value of DNMT1 and PR-B methylation as prediction tools for endometriosis with an LBW history, we performed an ROC curve analysis (Figure 1). The area under curve (AOC) of DNMT1 of 0.660 suggests a weak discriminatory ability, and the AOC of PR-B of 0.735 suggests an acceptable discriminatory ability. This ROC curve also showed the optimal cut-off value for both variables. The cut-off value of DNMT1 was ≤23.79, reaching 74.1% sensitivity, 68% specificity, and an RR of 2.45. Meanwhile, the cut-off value of PR-B methylation was ≤37.18, with 77.8% sensitivity, 80% specificity, and an RR of 3.5. The cut-off value of both the DNMT1 expression and PR-B methylation were moderately correlated with the endometriosis incidence (r = 0.422 and r = 0.577, respectively) (Table 3).

Furthermore, to determine the relationship between the simultaneous effect of the DNMT1 expression and PR-B methylation cut-off values on the incidence of endometriosis with LBW, a multiple logistic regression analysis was performed. From the analysis, we found that the most dominant factor associated with the incidence of endometriosis with LBW is PR-B (OR 10.40, 95% CI 3.24–33.4, R^2^ = 45.8%).

## 4. Discussion

Endometriosis is a common gynecological disorder with an enigmatic etiopathogenesis. Although it has been suggested that endometriosis is a hormonal, genetic disease and an inflammatory disease, understanding of its etiopathogenesis is still inadequate [3,18]. Studies trying to unravel the etiopathogenesis of endometriosis are still ongoing.

In the last decade, many studies have attempted to prove that endometriosis may be an epigenetic disease [10,11,18]. This is based on the suspicion that there are hormonal and immunological abnormalities in endometriosis. Epigenetic theory also appears to have better explanatory power of the disease. There is evidence regarding various epigenetic aberrations in the pathogenesis and pathophysiology of endometriosis. The implication of this finding is that endometriosis can then be treated by correcting epigenetic aberrations through pharmacological means [18]. DNA methylation markers are also known to be useful for endometriosis diagnosis and prognosis determination. Understanding the epigenetic changes and gene expression in endometriosis pathogenesis may hopefully lead to interventions that prevent the development of endometriosis in later life [18].

The influence of the intrauterine environment on the risk of endometriosis is still controversial. Several studies have tried to further reveal the correlation between several risks that are thought to somewhat trigger the incidence of endometriosis, including a history of LBW. Changes in fetal programming during the decidualization phase in intrauterine life may lead to epigenetic changes [5,6,7,8,9].

Epigenetic changes in fetal programming that occur in LBW babies are most likely influenced by intrinsic factors, namely a bad intrauterine environment due to malnutrition. This malnutrition state forces the fetal body to enter an amino acid (methionine) and lactate catabolism pathway. Methionine is the main donor to DNA and changes in methionine catabolism may result in a series of epigenetic changes [5,8,9].

In this study, 51.9% of women with endometriosis had a history of LBW, while 43.3% of non-endometriosis patients had a history of LBW. Our study also showed that women with a history of LBW had a 1.41-times-higher risk of endometriosis (95% CI 0.57–3.49) when compared to women with a history of normal birth weight, although the relationship showed no significant difference (*p* > 0.05). This result is in accordance with previous studies conducted by Borghese, et al., 2015 and Benagiano, et al. which strongly suggested that LBW is associated with a higher risk of developing deep endometriosis in adulthood [19,20]. Likewise, Brosens, 2012 and Kobayashi, 2014 stated that women with endometriosis have a higher risk of giving birth to LBW babies [6,21]. Vanuccinni, et al. 2016 also described the association of women with endometriosis with various pregnancy complications, including giving birth to LBW babies, PCOS, DM, hypertension, and preeclampsia [7].

DNA methylation is one interesting epigenetic mechanism. Increasing aberrant DNA methylation levels are associated with imprinting disorders in the prenatal ontogenetic processes of female reproductive organs. In endometriosis, the promoter region of PR-B is hypermethylated. This PR-B hypermethylation stimulates a feedback response that causes a down-regulation of progesterone receptor B that eventually leads to progesterone resistance [10,11].

This study showed that there is significantly more PR-B gene promoter DNA methylation occurring in the endometriosis group compared to the control group (*p* < 0.001). This state of PR-B gene promoter hypermethylation reduces PR-B receptor expression and interferes with the progesterone pathway in endometrial cells or tissues. Disturbances in PR-B methylation levels have also occurred in cases of endometrial cancer [22]. A study by Jichan et al. showed that the administration of the demethylating agent 5-aza-deoxycytidine (ADC) can restore PR-B expression in all endometrial cell layers [23]. This proves that PR-B gene inactivation occurred through DNA methylation at the PR-B promoter [10].

Methylation is known to be mediated by DNA methyltransferase-1 (DNMT1), an enzyme that adds a methyl group (CH3) to the cytosine base (CpG island) in the promoter region. Increased expression of DNMT1 is thought to be a factor contributing to high methylation levels [10,11]. Our study showed that DNMT1 expression in endometrial eutopic tissue in the menstrual blood of women with endometriosis was significantly higher than in that of the control group (*p* < 0.001). This finding is in line with the study by Wu, et al. 2007 which showed that the expression of DNMT1, DNMT3A, and DNMT3B were higher in the ectopic and eutopic endometrial tissue of women with endometriosis compared to controls. Mostly, Wu’s study of ectopic and eutopic endometrial samples were taken during the proliferative phase of the menstrual cycle, and only one sample each was taken during the menstrual phase out of 17 samples of the endometriosis group and 8 samples of the control group [24].

Zubrzycka et al. suggested that DNMT expression changes throughout menstrual phases [25]. Studies by Yamagata, et al., 2007 and Vincent LZ, et al., 2011 both showed that the expression of DNMT1, DNMT3A, and DNMT3B was higher in the proliferative phase compared to the secretory phase (*p* < 0.05) [26,27]. This is also in accordance with the study by Jjingo, et al., 2004 that showed high DNA methylation occurring during the proliferation phase and menstruation phase, but reduced DNA methylation during the secretory phase [28]. The higher DNMT1 expression during proliferative phase might be explained by the higher steroid levels during this phase. In this study, the samples were taken from menstrual blood. This might explain the significantly higher DNMT1 expression in the endometriosis group compared to the non-endometriosis group (*p* < 0.001).

Although not significant, our study showed a negative correlation between DNMT1 expression and PR-B expression among endometriosis women with LBW history. The increased expression of DNMT1, a mediator of promoter gene PR-B hypermethylation, suppresses the PR-B receptor expression and causes progesterone resistance. Brosens, et al., 2012 suggested that if the relationship between PR-B gene promoter hypermethylation and progesterone resistance could be harmonized, the PR-B gene promoter hypermethylation level could be regarded as a biological marker of progesterone resistance [6]. Disease prognosis can also be ascertained by determining the methylation level of the PR-B gene promoter in patients with endometriosis [16]. In this study, the most dominant factor associated with the incidence of endometriosis with LBW was PR-B methylation with an OR of 10.40 and an R2 magnitude of 45.8%.

In normal endometrial tissue, progesterone exerts an antiestrogenic effect, namely by stimulating 17βHSD type 2, which catalyzes the conversion of estradiol (E2), which has a strong biological effect, to estrone (E1), which has a weaker biological effect. However, in endometriosis, with progesterone resistance occurring, the level of 17βHSD type 2 decreases while estradiol levels remain high. Estrogen itself stimulates the growth of endometriosis, while progesterone inhibits the occurrence of endometriosis. Therefore, it can be concluded that progesterone resistance in endometriosis leads to increasing growth of endometrial tissue [12,13]. Furthermore, progesterone resistance and the uncontrolled increase of estradiol (E2) levels in endometrial tissue initiate the increase of one angiogenic factor, namely VEGF, that causes angiogenesis and neurogenesis in endometriosis. All of the above mechanisms lead to the development and progression of endometriosis (Figure 2) [4,14].

Our study found that the VEGF expression of eutopic endometrial cells from the menstrual blood of women with endometriosis was higher than that of non-endometriosis women. This finding is in line with a review that mentioned higher VEGF expression in the late secretory phase eutopic endometrium of endometriosis patients. VEGF produced by epithelial and stromal eutopic endometrial cells, especially the epithelium that is carried during retrograde menstruation, is thought to play a role in the survival of endometrial cells at ectopic sites in the early stages of implantation. After implantation, this triggers an inflammatory reaction and the movement of mast cells and macrophages to the lesion site which then releases VEGF and results in the continuation of angiogenesis [10].

Roskoski, et al. wrote in their literature review that in endometriosis there is an upregulation of multiple proangiogenic factors that causes a shift in the balance of pro and antiangiogenic factors. The most potent angiogenic factor is VEGF and the predominant proangiogenic factor in eutopic endometrium is VEGF A. VEGF A functions to inhibit apoptosis, stimulate mitogenesis, increase endometrial cell migration, and increase vascular permeability, thereby facilitating the formation of new blood vessels. These new blood vessels supply the nutritional and metabolic needs of the ectopic endometrial cells so that they survive in the peritoneal cavity. VEGF A, which is carried during retrograde menstruation, is thought to play a role in the early stages of endometrial cell implantation in ectopic sites. The VEGF A-mediated angiogenesis process is a key factor for the formation of vascular-rich red lesions that are early-stage lesions of peritoneal implantation before the emergence of the inflammatory response by macrophages and mast cells [29].

To evaluate the potential value of DNMT1 and PR-B methylation as prediction tools of endometriosis with LBW history, we performed an ROC curve analysis. We also performed a multiple logistic regression analysis and found that the most dominant factor associated with the incidence of endometriosis with LBW is PR-B (OR 10.40, 95% CI 3.24–33.4, R^2^ = 45.8). The cut-off value of PR-B methylation was ≤37.18%, with sensitivity 77.8%, specificity 80%, and RR = 3.5. Our study also proved that non-invasively taken endometrial cells, namely menstrual blood, can be isolated and used for research purposes like unraveling endometriosis etiopathogenesis, helping in disease diagnosis, and aiding in endometriosis therapy.

The questionnaire used in this study to gain biographical data, perinatal history, and menstrual history has not been validated, which could affect the accuracy of clinical data. The endometrial tissue obtained in this study was taken from menstrual blood in the posterior fornices of the patients. Even though this approach is less invasive and more comfortable for the patients, it could affect the purity of endometrial cells obtained as the sample. Epigenetic changes such as methylation could potentially be altered by other social or clinical factors other than an LBW history. Those factors were not considered in this study, since we emphasized the epigenetic level, but they could pose as confounding factors affecting the interpretation of our findings. This study also enrolled Indonesian women as the study subjects with their distinct genetic makeup and this could potentially affect the generalizability of the findings to other regions. These are the limitations of our study.

## 5. Conclusions

In conclusion, we have demonstrated that endometriosis is associated with PR-B gene promoter hypermethylation, decreased PR-B expression, and increased DNMT1 and VEGF expression. Furthermore, we found that PR-B methylation is the most dominant factor affecting endometriosis incidence. These perspectives on endometriosis epigenetics may provide new preventive efforts against endometriosis.

## Figures and Tables

**Figure 1 diagnostics-13-02085-f001:**
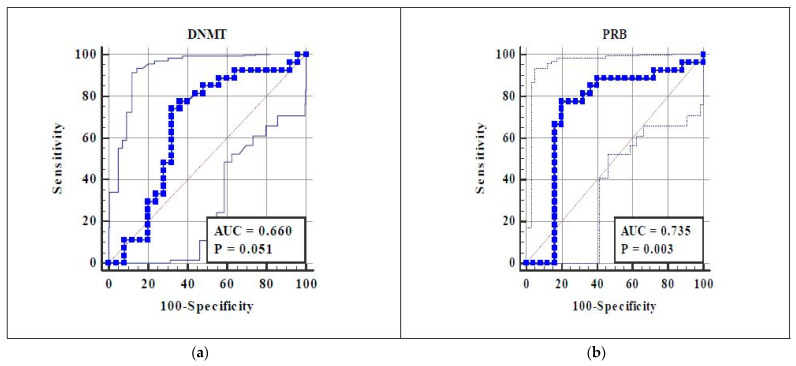
ROC curve of (**a**) DNMT1 expression and (**b**) PR-B methylation as predictors of endometriosis with an LBW history. The ROC curve is depicted as a blue line, and its standard error is depicted as the black line.

**Figure 2 diagnostics-13-02085-f002:**
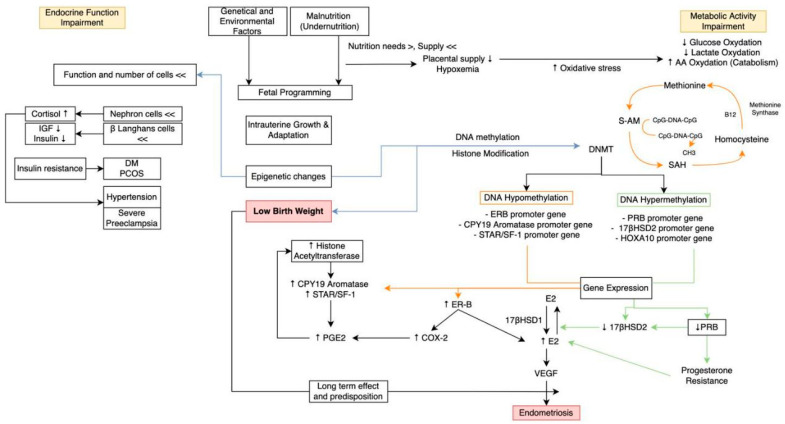
Pathomechanism of endometriosis through the “Malnutrition—Fetal Programming—Epigenetic—Low Birth Weight” pathway.

**Table 1 diagnostics-13-02085-t001:** Baseline characteristics of endometriosis and non-endometriosis patients.

Characteristic	Group		*p* Value *
	Endometriosis (*n* = 52)	Non-Endometriosis (*n* = 30)	
Age (years):			0.983
Mean (SD)	34.3 (4.9)	34.3 (5.6)	
Range	25–45	24–43	
Education:			0.702
High School	22 (42.3%)	14 (46.7%)	
Diploma/Undergraduate	30 (57.7%)	16 (53.3%)	
Parity:			0.360
0	19 (36.5%)	8 (26.7%)	
1–3	33 (63.5%)	22 (73.3%)	
BMI (kg/m^2^):			0.506
Median	22.77	21.78	
Range	17.21–28.89	17.67–35.03	

Note: * age was tested with *t* test; BMI with Mann–Whitney test; education and parity with chi-square test.

**Table 2 diagnostics-13-02085-t002:** Differences in PR-B methylation, DNMT1 expression, PR-B and VEGF in the four study groups.

Group					
Variable	Endometriosis with LBW (*n* = 27)	Endometriosis without LBW (*n* = 25)	Non-Endometriosis with LBW (*n* = 13)	Non-Endometriosis without LBW (*n* = 17)	*p* Value *
1. PR-B Methylation (%)					<0.001
Median	17.67 ^a^	16.33	3.23	3.67	
Range	(11.0–21.7)	(11.33–20.66)	(1.33–5.33)	(1.33–9.67)	
2. DNMT1:					<0.001
Median	13.33 ^a^	10.37	4.86	7.16	
Range	(6.67–16.58)	(6.95–22.23)	(1.00–17.07)	(1.18–20.36)	
3. PR-B:					<0.001
Median	5.67 ^a^	7.43	9.43	8.89	
Range	(1.15–9.63)	(1.33–10.03)	(6.58–11.99)	(6.58–13.23)	
4. VEGF:					<0.001
Mean (SD)	11.33 ^a^	9.32	6.72	7.53	
Range	(5.75–32.67)	(5.67–20.67)	(5.63–8.12)	(6.21–8.35)	

Note: * Kruskal–Wallis Test; ^a^ has a statistically significant difference compared with other groups (*p* < 0.05).

**Table 3 diagnostics-13-02085-t003:** Cut-off values for DNMT1 and PR-B methylation as predictors of LBW in endometriosis subjects.

Cut-Off Value	Endometriosis		*p* Value *	Notes
	With LBW	Without LBW		
	(*n* = 27)	(*n* = 25)		
DNMT1:			0.002	Sensitivity = 74.1%
				Specificity = 68%
≤23.79%	20 (74.1%)	8 (32.0%)		RR (CI 95%):
>23.79%	7 (25.9%)	17 (68.0%)		2.45 (1.28–4.77)
				Phi (correlation) = 0.422
2. PR-B:			<0.001	Sensitivity = 77.8%
				Specificity = 80%
>37.18%	21 (77.8%)	5 (20.0%)		RR (CI 95%):
≤37.18%	6 (22.2%)	20 (80.0%)		3.50 (1.69–7.24)
				Phi (correlation) = 0.577

Notes: * Chi-square test; RR (CI 95%).

## Data Availability

Raw dataset used for this manuscript are available from the corresponding authors on reasonable request.

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
