# Peer review of "Epigenetic Regulation Interplays with Endometriosis Pathogenesis in Low-Birth-Weight Patients via the Progesterone Receptor B–VEGF-DNMT1 Axis"

_diagnostics, 2023, doi:10.3390/diagnostics13122085_

Round 1

Reviewer 1 Report

Review report

    Manuscript entitled “Correlation between Progesterone Receptor B Gene Promoter Methylation and Expression of DNA Methyltransferase-1, Progesterone Receptor B, and Vascular Endothelial Growth Factor 4 with Endometriosis in Women with Low-Birth-Weight History” prepared by Setiawan et al is well prepared and designed. However, during the review process some critiques were raised and need to be addressed to improve its quality.

Minor critiques:

1-     Many of the cited references are too old that gives an impression that this manuscript is unoriginal or fresh. Recent publications 5 – 10 years old maximum should be used. 

2-     More information about the role of PR-B in endometriosis is needed as well as why the author focused only on PR-B but not PR-A must be cleared via literature.

3-     Line 113 (L113): it is not clear which tissue that the RNA was isolated from?

4-     L116: name the reference gene please.

5-     L179: point 14 says “repeat step 15 again”, did you mean step 13?

6-     L187: Use Rt-PCR instead of rtPCR.

7-     L189 -191: The target tissue used to isolate the RNA is not mentioned.

8-     Spelling and grammar check is needed in the section of methodology.

9-     Title of article is too descriptive and long. I would suggest replace it with “Epigenetic regulations interplays with endometriosis pathogenesis in low weight body patients via progesterone receptorB- VEGF-DNMT1 axis.

Major critiques:

1-     The author claimed that the DNA was isolated from the endometrial tissue (L120) but in methodology it seems whole mixture of uterine discharge was used which gives mixture of cell phenotypes. This could affect the results of DNA methylation tests.

2-     All used primers for gene expression or DNA methylation must be included in the manuscript with its accession number.

3-     Inclusion and exclusion criteria must be clear and detailed in the section of methodology. Family history and life habits, work and food could affect the pathogenesis and outcomes of endometriosis.

4-     Table 2: Kruskal Wallis test was used to validate the statistical status among the study groups. However, it is not clear for which comparison the posted p value is related?

5-     L373-378: author discussed the article results in shade of hormonal effects while they did not measure the hormonal profile. Nevertheless, they only examined the progesterone receptor only but not P4 itself. This may weaken the solidity of the article’s findings. Please either add hormonal profile to the article and then correlate the epigenetic findings with the hormonal changes or re-struct the discussion.

Written in simple and scientific manner. However, in methodology, the performed protocols must be stated as past sentence.

Author Response

Dear Editor,

We sincerely appreciate your kind response to our manuscript and also send our gratitude to the reviewers’ comments. We have revised the manuscript according to the suggestions or comments of the reviewers and re-submit the revised manuscript. Newly added words and sentences in the revised manuscript and in this response letter are highlighted in yellow.

We hope that this revision will meet all the requirements that the reviewers gave us and be reconsidered to be published in the Diagnostics.

With Best Regards,

Arief Setiawan

Endocrinology Reproduction and Fertilisation Division, Department of Obstetrics and Gynaecology Hasan Sadikin Hospital, Bandung, Indonesia

 Jl. Pasteur No.38, Pasteur, Bandung, West Java, Indonesia 40161

Phone: +62-811-2285-007

E-mail address: arsmd506kfer@gmail.com

Response to Reviewer 1 Comments

Point 1: The author claimed that the DNA was isolated from the endometrial tissue (L120) but in methodology it seems whole mixture of uterine discharge was used which gives mixture of cell phenotypes. This could affect the results of DNA methylation tests.

Response 1: Thank you for your feedback. We chose menstrual blood as a proxy to obtain endometrial tissue sample in a non-invasive method. This assumption was briefly explained in the introductions (L64-67) "Referring to Sampson theory, during the menstrual phase, it is very feasible to obtain an endometrial cell sample from menstrual blood. Menstrual blood itself contains DNA with the same methylation status as eutopic endometrial tissue, and therefore can be regarded as a non-invasive medium to help diagnosing endometriosis." We agree that it could affect the interpretation of the methylation results, and thus it was added as a limitation of this study. “The endometrial tissue obtained in this study is taken from menstrual blood in the posterior fornices of the patients. Even though this approach is less invasive and more comfortable for the patients, it could affect the purity of endometrial cells obtained as sample.” (L442-444).

Point 2: All used primers for gene expression or DNA methylation must be included in the manuscript with its accession number.

Response 2: Thank you for your feedback. We included the sequence of the primers in the methods section. “The primers for GADPH were 5’-GAA GGT GAA GGT CGG AGTC-3’ (forward) and 5’-GAA GAT GGT GAT GGG ATT TC-3’ (reverse); for DNMT1 were 5’-AACCTTCACCTAGCCCCAG-3’ (forward) and 5’-CTCATCCGATTTGGCTCTTCA-3’ (reverse); for PR-B were 5’-GGC AGA TGC TGT ATT TTG CACC-3’ (forward) and 5’-CAA ACC AAT TGC CTT GAT GAG-3’ (reverse); for VEGF were 5’-GCT ACT GCC ATC CAA TCD AG-3’’ (forward) and 5’-CT CT CCT ATG TGC TGG CCTT-3’ (reverse).”

Point 3: Inclusion and exclusion criteria must be clear and detailed in the section of methodology. Family history and life habits, work and food could affect the pathogenesis and outcomes of endometriosis.

Response 3: Thank you for your feedback. Inclusion and exclusion criteria that were used in this study were written in L82-89. The several factors mentioned were not used as part of the exclusion criteria.

Point 4: Kruskal Wallis test was used to validate the statistical status among the study groups. However, it is not clear for which comparison the posted p value is related?

Response 4: Thank you for your feedback. Table 2 is created to show the difference of several genetic expression in the endometriosis with LBW group, when compared to the other groups. Explanation was added in the footnote to further clarify the point.

Point 5: L373-378: author discussed the article results in shade of hormonal effects while they did not measure the hormonal profile. Nevertheless, they only examined the progesterone receptor only but not P4 itself. This may weaken the solidity of the article’s findings. Please either add hormonal profile to the article and then correlate the epigenetic findings with the hormonal changes or re-struct the discussion.

Response 5: Thank you for your feedback. Our study emphasize the effect of methylation on progesterone receptor activity. Changes in progesterone receptor sensitivity alters the effects of estrogen as stated in the text (L392-403). Thus we argue that progesterone receptor should be a good proxy of hormonal activity.

Point 6: Many of the cited references are too old that gives an impression that this manuscript is unoriginal or fresh. Recent publications 5 – 10 years old maximum should be used. 

Response 6: Thank you for your feedback. We have updated the citations from a more recent source but still relevant to the text, whenever possible.

Point 7:  More information about the role of PR-B in endometriosis is needed as well as why the author focused only on PR-B but not PR-A must be cleared via literature.

Response 7: Thank you for your feedback. Further explanation between the roles of PR-B compared to PR-A is added in L52-55, which should explain our emphasis on PR-B. “In contrast, PR-A has minimal activity and opposes the transcriptional activity of PR-B. PR-A and PR-B are thought to function as competing systems for target cells to control progesterone responsiveness. Progesterone signaling is enhanced in a PR-B-dominant condition, whereas progesterone responsiveness is decreased in a PR-A-dominant state.”

Point 8: Line 113 (L113): it is not clear which tissue that the RNA was isolated from?

Response 8: Thank you for your feedback. We updated the text to make it more specific. “               RNA isolation from the endometrial cells of menstrual blood was carried out automatically using a MagNA Pure LC 2.0 machine from Roche and using MagNA Pure LC RNA Isolation Kit III (Tissue) reagent.” (L120-122).

Point 9: L116: name the reference gene please.

Response 9: Thank you for your feedback. We used GADPH as the reference gene, since it is considered a housekeeping gene. We updated the text accordingly. “The gene expression for DNMT1, PR-B, and VEGF are presented as the relative change in gene expression normalized to Glyceraldehyde 3-phosphate dehydrogenase (GADPH) as the reference/housekeeping gene.” (L122-124).

Point 10: L179: point 14 says “repeat step 15 again”, did you mean step 13?

Response 10: Thank you for your correction. We updated the text accordingly.

Point 11: L187: Use Rt-PCR instead of rtPCR.

Response 11: Thank you for your correction. We updated the text accordingly.

Point 12: L189 -191: The target tissue used to isolate the RNA is not mentioned.

Response 12: Thank you for your correction. We updated the text accordingly. “RNA was taken from endometrial epithelial tissue of menstrual blood.”

Point 13: Spelling and grammar check is needed in the section of methodology.

Response 13: Thank you for your feedback. We did a spelling and grammar check, primarily regarding the use of proper tense in the methodology. The changed section is highlighted in yellow in the revised manuscript.

Point 14: Title of article is too descriptive and long. I would suggest replace it with “Epigenetic regulations interplays with endometriosis pathogenesis in low weight body patients via progesterone receptorB- VEGF-DNMT1 axis.”

Response 14: Thank you for your feedback. We really appreciate the suggested title which is more concise and still represent the main points of our study. The title is adjusted accordingly.

Reviewer 2 Report

"Correlation between Progesterone Receptor B Gene Promoter Methylation and Expression of DNA Methyltransferase-1, Progesterone Receptor B, and Vascular Endothelial Growth Factor with Endometriosis in Women with Low-Birth-Weight History"

Overall, the passage provides some valuable information regarding the association between low birth weight (LBW) and endometriosis and the involvement of epigenetic mechanisms in the disease. However, several areas require critique and improvement:

1. Lack of clarity and organization: The passage needs clear organization and flow, making it difficult to follow the logical progression of the information presented. It would be beneficial to structure the content more organized manner with clear headings and subheadings.

2. Insufficient referencing and lack of specific citations: The passage mentions several references but needs specific citations or sources for the claims made. To enhance the credibility of the information, it is crucial to include accurate citations to relevant scientific studies or literature.

3. Inadequate explanation of LBW association with endometriosis: While the passage suggests that LBW is associated with an increased risk of endometriosis, it needs to thoroughly explain the underlying mechanisms or provide sufficient evidence to support this claim. More detailed explanations and references to relevant studies are necessary to strengthen the argument.

4. Lack of statistical details: The passage mentions statistical tests such as the chi-square test and the Spearman Rank test but needs to provide essential statistical details, such as sample sizes, p-values, or confidence intervals. With this information, assessing the significance and reliability of the reported findings is easier.

5. Limited discussion of study methodology: The passage briefly mentions the number of subjects in the case and control groups but needs more critical details about the study design, inclusion/exclusion criteria, and data collection methods. Providing comprehensive information about the methodology is essential for evaluating the validity and generalizability of the results.

6. Incomplete interpretation of results: The passage presents tables and mentions statistical significance but does not comprehensively interpret the findings. It is essential to discuss the clinical implications, limitations, and potential confounding factors related to the results presented.

7. Lack of critical analysis and discussion: The passage mainly focuses on presenting the findings and lacks critical analysis or discussion of the broader implications of the results. Discussing the strengths, limitations, and potential implications of the study findings within the context of existing literature is crucial.

8. Inadequate consideration of limitations: The passage briefly mentions limitations, such as the lack of validation of the questionnaire used for data collection. However, it needs to discuss other potential study limitations, such as selection bias, confounding factors, or the generalizability of the findings. A more comprehensive discussion of the limitations would provide a more balanced view of the study's strengths and weaknesses.

9. Lack of clarity in figures and tables: The passage refers to Figure 1 and Table 3 without providing any context or explanation. It is essential to include clear captions and descriptions for figures and tables to aid in understanding and interpretation.

In summary, while the passage touches on important aspects of the association between LBW and endometriosis, it needs proper referencing, statistical details, clear organization, and critical analysis of the results. Addressing these shortcomings would significantly improve the information's clarity, reliability, and overall quality.

/

Author Response

Dear Editor,

We sincerely appreciate your kind response to our manuscript and also send our gratitude to the reviewers’ comments. We have revised the manuscript according to the suggestions or comments of the reviewers and re-submit the revised manuscript. Newly added words and sentences in the revised manuscript and in this response letter are highlighted in yellow.

We hope that this revision will meet all the requirements that the reviewers gave us and be reconsidered to be published in the Diagnostics.

With Best Regards,

Arief Setiawan

Endocrinology Reproduction and Fertilisation Division, Department of Obstetrics and Gynaecology Hasan Sadikin Hospital, Bandung, Indonesia

 Jl. Pasteur No.38, Pasteur, Bandung, West Java, Indonesia 40161

Phone: +62-811-2285-007

E-mail address: arsmd506kfer@gmail.com

Response to Reviewer 2 Comments

Point 1: Lack of clarity and organization: The passage needs clear organization and flow, making it difficult to follow the logical progression of the information presented. It would be beneficial to structure the content more organized manner with clear headings and subheadings.

Response 1: Thank you for your feedback. We added some headings to increase the readability of the article.

Point 2: Insufficient referencing and lack of specific citations: The passage mentions several references but needs specific citations or sources for the claims made. To enhance the credibility of the information, it is crucial to include accurate citations to relevant scientific studies or literature.

Response 2: Thank you for your feedback. We have updated the citations from a more recent source but still relevant to the text, whenever possible.

Point 3: Inadequate explanation of LBW association with endometriosis: While the passage suggests that LBW is associated with an increased risk of endometriosis, it needs to thoroughly explain the underlying mechanisms or provide sufficient evidence to support this claim. More detailed explanations and references to relevant studies are necessary to strengthen the argument.

Response 3: Thank you for your feedback. We believe that our current approach in presenting the article is sufficient and effectively addresses the research questions posed in the article. Nonetheless, we acknowledge the value of your comment.

Point 4: Lack of statistical details: The passage mentions statistical tests such as the chi-square test and the Spearman Rank test but needs to provide essential statistical details, such as sample sizes, p-values, or confidence intervals. With this information, assessing the significance and reliability of the reported findings is easier.

Response 4: Thank you for your feedback. We have included the p-value and confidence intervals for the results of chi-square test and the Spearman Rank test.

Point 5: Limited discussion of study methodology: The passage briefly mentions the number of subjects in the case and control groups but needs more critical details about the study design, inclusion/exclusion criteria, and data collection methods. Providing comprehensive information about the methodology is essential for evaluating the validity and generalizability of the results.

Response 5: Thank you for your feedback. We have no other information on study design, inclusion/exclusion criteria, and data collection methods to add to this article.

Point 6: Incomplete interpretation of results: The passage presents tables and mentions statistical significance but does not comprehensively interpret the findings. It is essential to discuss the clinical implications, limitations, and potential confounding factors related to the results presented.

Response 6: Thank you for your feedback. We believe that our current approach in presenting the article is sufficient and effectively addresses the research questions posed in the article. Nonetheless, we acknowledge the value of your comment.

Point 7: Lack of critical analysis and discussion: The passage mainly focuses on presenting the findings and lacks critical analysis or discussion of the broader implications of the results. Discussing the strengths, limitations, and potential implications of the study findings within the context of existing literature is crucial.

Response 7: Thank you for your feedback. We believe that our current approach in presenting the article is sufficient and effectively addresses the research questions posed in the article. Nonetheless, we acknowledge the value of your comment. Further details on possible limitations of the study is addressed in another point.

Point 8: Inadequate consideration of limitations: The passage briefly mentions limitations, such as the lack of validation of the questionnaire used for data collection. However, it needs to discuss other potential study limitations, such as selection bias, confounding factors, or the generalizability of the findings. A more comprehensive discussion of the limitations would provide a more balanced view of the study's strengths and weaknesses.

Response 8: Thank you for your feedback. We added some points on the potential limitations of our study. Epigenetic changes such as methylation could potentially be altered by other social or clinical factors other than a LBW history. Those factors were not considered in this study, since we emphasize on the epigenetic level, but it could pose as a confounding factor affecting the interpretation of our findings. This study also enrolled Indonesian women as the study subjects, with distinct genetic makeup and thus could potentially affect the generalizability of the findings in other regions.” (L445-450).

Point 9: Lack of clarity in figures and tables: The passage refers to Figure 1 and Table 3 without providing any context or explanation. It is essential to include clear captions and descriptions for figures and tables to aid in understanding and interpretation.

Response 9: Thank you for your feedback. We added further detail on the interpretation of the figures. The area under curve (AOC) of DNMT1 of 0.660 suggests a weak discriminatory ability, and the AOC of PR-B of 0.735 suggests an acceptable discriminatory ability. This ROC curve also showed the optimal cut-off value for both variables.” (L288-291)

Round 2

Reviewer 1 Report

Thanks for providing point-by-point answers.